# Proof-of-Concept of Spent Mushrooms Compost Torrefaction—Studying the Process Kinetics and the Influence of Temperature and Duration on the Calorific Value of the Produced Biocoal

**Ewa Syguła [1], Jacek A. Koziel [2] and Andrzej Białowiec [1,2,]***

[1] Faculty of Life Sciences and Technology, Institute of Agricultural Engineering, Wrocław University of Environmental and Life Sciences, 37/41 Chełmońskiego Str., 51-630 Wroclaw, Poland

[2] Department of Agricultural and Biosystems Engineering, Iowa State University, Ames, IA 50011, USA

* Correspondence: andrzej.bialowiec@upwr.edu.pl; Tel.: +48-71-320-5973

**Abstract:** Poland, being the 3rd largest and growing producer of mushrooms in the world, generates almost 25% of the total European production. The generation rate of waste mushroom spent compost (*MSC*) amounts to 5 kg per 1 kg of mushrooms produced. We proposed the *MSC* treatment via torrefaction for the production of solid fuel—biocoal. In this research, we examined the *MSC* torrefaction kinetics using thermogravimetric analyses (TGA) and we tested the influence of torrefaction temperature within the range from 200 to 300 °C and treatment time lasting from 20 to 60 min on the resulting biocoal's (fuel) properties. The estimated value of the torrefaction activation energy of *MSC* was 22.3 kJ mol$^{-1}$. The highest calorific value = 17.9 MJ kg$^{-1}$ d.m. was found for 280 °C (60 min torrefaction time). A significant ($p < 0.05$) influence of torrefaction temperature on *HHV* increase within the same group of torrefaction duration, i.e., 20, 40, or 60 min, was observed. The torrefaction duration significantly ($p < 0.05$) increased the *HHV* for 220 °C and decreased *HHV* for 300 °C. The highest mass yield (98.5%) was found for 220 °C (60 min), while the highest energy yield was found for 280 °C (60 min). In addition, estimations of the biocoal recirculation rate to maintain the heat self-sufficiency of *MSC* torrefaction were made. The net quantity of biocoal (torrefied *MSC*; 65.3% moisture content) and the 280 °C (60 min) torrefaction variant was used. The initial mass and energy balance showed that *MSC* torrefaction might be feasible and self-sufficient for heat when ~43.6% of produced biocoal is recirculated to supply the heat for torrefaction. Thus, we have shown a concept for an alternative utilization of abundant biowaste (*MSC*). This research provides a basis for alternative use of an abundant biowaste and can help charting improved, sustainable mushroom production.

**Keywords:** waste to energy; waste to carbon; mushroom spent compost; biocoal; torrefaction; activation energy; fuel properties; circular economy

## 1. Introduction

The production of mushrooms in Poland is growing rapidly. During 2006–2018, the production of mushrooms increased from 196 thousand Mg·year$^{-1}$ to 325 thousand Mg·year$^{-1}$, which currently accounts for 24.15% of the total production in the European Union (EU) [1,2]. Poland is the largest grower of mushrooms in Europe and the third in the world. One of the abundant wastes arising during the cultivation of mushrooms is a worn-out substrate—mushroom spent compost (*MSC*)—which accounts to ~5 kg of *MSC* per 1 kg of produced mushrooms [3]. This amounts to 5.6 and 1.6 million Mg·year$^{-1}$ of *MSC*, in the EU and Poland, respectively [1,4].

The sustainable management of *MSC* is an important issue for the industry. The *MSC* is currently considered as an 'industrial waste' (020199 code, according to European Waste Classification), which is classified as wastes from agriculture [5]. Potential waste treatment methods depend on *MSC* properties. For example, a white mushroom (*Agaricus bisporus*) is grown on a bedding consisting mainly of straw and poultry manure. The manure also contains gypsum, urea, peat, coconut, and soy proteins [6]. The mushroom production cycle takes about 6 to 8 weeks on this bedding with three harvestings of mushrooms. After this period, the *MSC* cannot be regenerated and used again for production. It is a waste which needs to be treated [7]. The *MSC* is a source of biogenic elements, such as C, N, P, K, and S (Table 1). The low level of heavy metals content allows the *MSC* to be considered as an organic fertilizer [8]. In addition, the *MSC* also contains a residual fraction of organic compounds, humic and fulvic acids, fraction decalcification, and bitumen [7]. An overview of the properties of the *MSC* is presented in Table 1.

**Table 1.** The properties of mushroom spent compost (*MSC*) given in the literature [1,6,9–12].

| Parameter | Unit | Values |
|---|---|---|
| Bulk density | kg·m$^3$ | 196.5 |
| pH | - | 6.23–7.15 |
| Moisture | % | 68.7–56.24 |
| Volatile solids | % d.m. | 53.97–65.18 |
| Ash | % d.m. | 22.43–38.3 |
| Higher Heating Value, *HHV* | MJ·kg$^{-1}$ d.m. | 4.06–12.98 |
| Lower Heating Value, *LHV* | MJ·kg$^{-1}$ d.m. | 2.03–10.17 |
| C | % d.m. | 27.8–40.73 |
| H | % d.m. | 3.40–4.86 |
| N | % d.m. | 1.80–6.00 |
| S | % d.m. | 2.12–2.99 |
| Cl | % d.m. | 0.1 |
| O | % d.m. | 18.94–26.99 |
| P | g·kg$^{-1}$ d.m. | 9.06–18.00 |
| K | g·kg$^{-1}$ d.m. | 15.2–20.00 |
| C/N | - | 12.3–18.0:1 |
| Ca | g·kg$^{-1}$ d.m. | 28–109 |
| Mg | g·kg$^{-1}$ d.m. | 3.6–18.0 |
| Na | g·kg$^{-1}$ d.m. | 1.60–1.68 |
| Lignin | % d.m. | 25 |
| Cellulose | % d.m. | 38 |
| Hemicellulose | % d.m. | 19 |
| Cu | mg·kg$^{-1}$ d.m. | 18.3–54.0 |
| Zn | mg·kg$^{-1}$ d.m. | 143.0–168.1 |
| Mn | mg·kg$^{-1}$ d.m. | 164.0–336.8 |
| Fe | mg·kg$^{-1}$ d.m. | 4.7–4494.5 |
| Mo | mg·kg$^{-1}$ d.m. | 1.51–2.13 |
| Al | mg·kg$^{-1}$ d.m. | 987 |
| B | mg·kg$^{-1}$ d.m. | 12.5–47.7 |
| Se | mg·kg$^{-1}$ d.m. | 2.25 |
| Li | mg·kg$^{-1}$ d.m. | 3.27 |
| Ti | mg·kg$^{-1}$ d.m. | 18.0 |
| Pb | mg·kg$^{-1}$ d.m. | 2.47–10.40 |
| Cd | mg·kg$^{-1}$ d.m. | 0.089–6.200 |
| Cr | mg·kg$^{-1}$ d.m. | 0.21–5.80 |
| Ni | mg·kg$^{-1}$ d.m. | 2.75–13.30 |

Thus far, various methods for *MSC* management have been developed and used. Those methods were aimed to obtain compost, bioethanol, biogas, enzyme lactase, xylooligosaccharides, or hydrogen. The most popular use of *MSC* is composting due to its organic nature and balanced ratio of C/N

12.3–18:1 (Table 1). The work of Kalembasa et al. [7] showed that compost from *MSC* could be an excellent fertilizer for improving soil structure. However, research to date shows some concerns as well [13]. For example, compost from *MSC* can be highly variable and does not always meet the legal criteria for a fertilizer [8]. This is due to the high variability of the compost parameters such as N, P, and K, biogenic elements, C/N ratio, pH, electrolytic conductivity, and the Ca, Na, Mg, Cu, Fe, Zn, Mn, Cr, Cd, Pb, and Ni content. The share and content of hemicellulose, cellulose, and lignin can also vary [14]. Thus, the variable *MSC* composition may cause technological problems of the process and heterogeneity of the obtained fertilizer quality.

Bioethanol production from *MSC* can be facilitated via hydrolysis with acids or bases, physical treatment using steam followed by fermentation. The difficulty comes from high lignin content. Steam explosion is used for delignification (the breakdown of lignin structures) into simpler cellulose. The literature shows that the highest potential of conversion of *MSC* to bioethanol requires 20 bar steam pressure [15]. Research on *MSC* hydrolysis with the addition of surfactants was published [16].

The *MSC* may also be a source of valuable substances. During the hydrothermal conversion of *MSC* rich in nutrients (e.g., proteins), production of xylooligosaccharides may be possible [17,18]. Thanks to the high purity of the obtained product, this process has industrial potential [15]. The *MSC* may also be an efficient source of lactase enzyme, which is widely used in paper, clothing, and food production [4].

Biogas production is also feasible. However, it has been found that this process is not as effective as fermentation of other biomass. Anaerobic fermentation of *MSC* generates ~122 $dm^3 \cdot kg^{-1}$ d.m. of biomethane, while corn silage can produce 320 $dm^3 \cdot kg^{-1}$ d.m. [19]. The biogas yield from other available substrates: cattle manure (324 ± 15.5 $dm^3 \cdot kg^{-1}$ d.m.), corn silage (653 ± 28.8 $dm^3 \cdot kg^{-1}$ d.m.), waste fruit and vegetables (678 ± 15.8 $dm^3 \cdot kg^{-1}$ d.m.), sugar beet pulp (634 ± 235 $dm^3 \cdot kg^{-1}$ d.m.), whey (736 ± 15.5 $dm^3 \cdot kg^{-1}$ d.m.) [20], shows that biogas production from *MSC* is not competitive enough. Another interesting use of *MSC* is the production of hydrogen, which has already been tested at the lab-scale. Two-step hydrolysis with acid and base resulted in 2.52 moles of $H_2 \cdot g$ $COD^{-1}$ [21].

Law regulations regarding the management of waste from mushroom production allow using *MSC* for energy purposes [22]. The 2009/28/WE (23 April 2009) [23] indicates the possibility of considering the *MSC* as biomass, and in consequence, the energy produced from *MSC* as energy from a renewable source. However, due to the high moisture, *MSC* is considered as fuel with low calorific value (Table 1) [24]. The *MSC*, with approximately 70% moisture, has an *LHV* of ~4.6 $MJ \cdot kg^{-1}$ d.m. [3]. *MSC*'s characteristics (Table 1) do not indicate high profitability of the incineration process [25].

We propose to convert the *MSC* into a more efficient solid fuel, according to the 'Waste to Carbon' approach [26] (Figure 1). One of the ways of *MSC* conversion is the process of torrefaction, otherwise known as high-temperature drying, low-temperature pyrolysis, or biomass 'roasting'. It consists of roasting the organic compounds of plant origin out of a substance [27]. Torrefaction is characterized by the range of temperatures 200–300 °C [28], while according to the act on renewable sources energy [29], the temperature range is 200–320 °C. The residence (process) time of torrefaction depends on the water or volatile content, as well as the type of reactor or substrate type [30]. The residence time usually does not exceed 60 min. The products of torrefaction are solid biocoal and gas ('torgas'). Torgas consists of nonflammable substances such as water or $CO_2$ and flammable substances such as CO, $CH_4$, or $H_2$ [31]. Depending on process parameters and properties of the substrate, a different ratio of flammable to noncombustible parts may be achieved [28]. The torgas is not an attractive product of torrefaction due to the high content of nonflammable substances [32,33].

The main product of the torrefaction, biocoal, according to Polish regulations, should have *LHV* not less than 21 $MJ \cdot kg^{-1}$ d.m. and the feedstock should be from solid biomass/waste of vegetable, animal, or biodegradable origin to be considered as a 'renewable source of energy' [29].

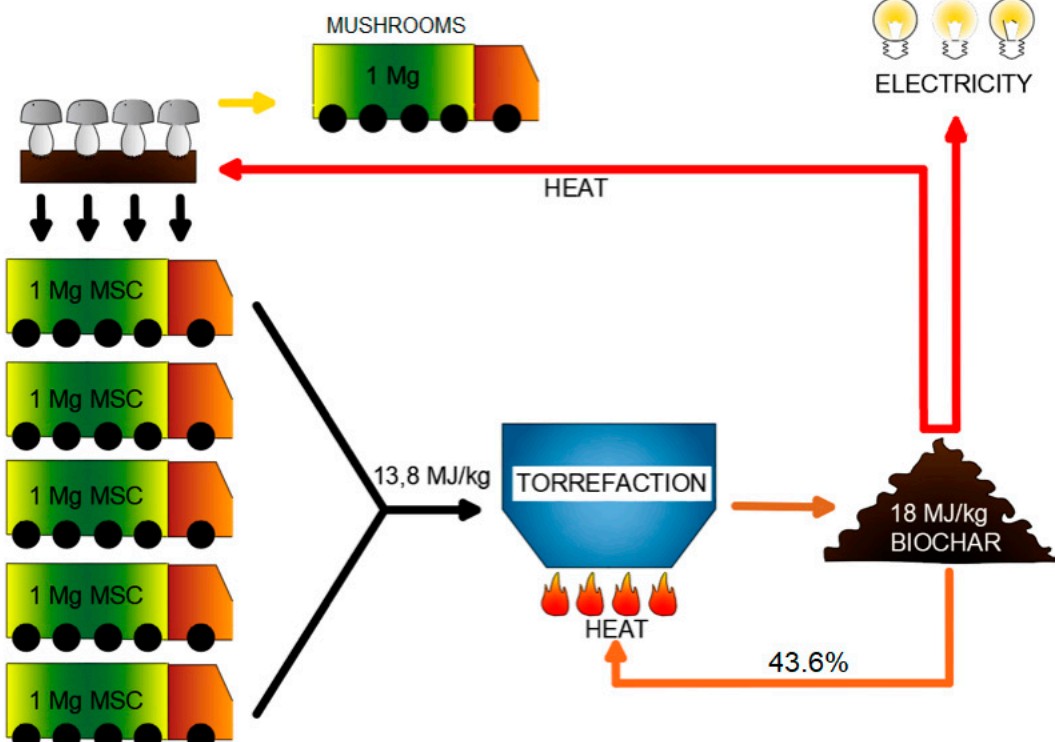

**Figure 1.** The proposed concept of valorizing mushroom spent compost (*MSC*) via torrefaction. Resulting biocoal can be renewable fuel, and the process of torrefaction may be self-sufficient for heat due to the reuse of a part of produced biocoal.

During the torrefaction process, the loss of mass, as well as the loss of chemical energy from raw material, is observed. However, the degree of mass loss is between 30 and 40%, while the energy content drop does not exceed 10%. Due to this difference, energy densification in solid fuels occurs [28]. However, the bulk density of the product is increasing, which positively affects the logistics of fuel [25] transport. Torrefaction increases the C content in the biocoal, which improves its calorific value. The product also becomes more homogenous. This affects the further thermal transformation of biocoal as a fuel. Because the abrasiveness is reduced, the mechanical grinding requires less energy compared to the raw substrate. There is also a possibility of pelletization of biocoal from torrefaction [34]. Additionally, the ratios of H/C and O/C are reduced, which improves the fuel properties of the biocoal in relation to the used substrate [35,36].

Torrefaction of waste is a growing research area. Examples include fractions of industrial and municipal wastes [27,37], sewage sludge [38], brewers' spent grain [39], Oxytree biomass [40] prunings, woodchips, olives waste, and Virginia mallow (considered as a potential energy crop) [22]. High-quality fuels can be obtained due to the torrefaction process. Therefore, it is necessary to carry out research on new substrates, aimed to optimize the torrefaction process to obtain the highest calorific value. For torrefaction optimization, it is necessary to determine the appropriate temperature and residence time for a given substrate to achieve the most beneficial fuel parameters [25,41].

The *MSC* is a new substrate which has not yet been tested for its conversion to solid fuel during the torrefaction process. The torrefaction of *MSC* and reusing part of the heat from biocoal combustion for torrefaction energy demand and *MSC* farm energy demand is at the early stage of technology readiness. Therefore, the aim of this study was the determination of the *MSC* torrefaction process kinetics parameters and the preliminary selection of technological parameters under which biocoal with the highest calorific value may be obtained. This article presents pioneering, novel research as a proof-of-concept of a new technology of *MSC* reuse, by conversion to high-quality solid fuel, which may be internally reused to achieve heat self-sufficiency (Figure 1).

## 2. Materials and Methods

### 2.1. The Properties of MSC

The samples of *MSC* originated from a mushroom farm in Gogołowo, Poland. According to Dudek et al. [39], *MSC* samples were first dried for 24 h in a WAMED lab dryer (KBC-65W, Warsaw, Poland) under the temperature of 105 °C. Then, dry *MSC* sample was ground through a 0.1 mm screen. For grinding, the laboratory knife mill TESTCHEM, model LMN-100 (Pszów, Poland), was used. The raw biomass of *MSC* was characterized by the determination of moisture content, volatile solids (*VS*) content, ash content, and higher heating value (*HHV*) (Table 2).

**Table 2.** The properties of *MSC* used for torrefaction.

| Parameter | Unit | Mean ± Standard Deviation |
|---|---|---|
| Moisture | % | 65.32 ± 0.05 |
| VS | % d.m. | 71.60 ± 2.31 |
| Ash | % d.m. | 28.40 ± 2.31 |
| HHV | $MJ \cdot kg^{-1}$ d.m. | 13.79 ± 0.50 |

### 2.2. MSC Torrefaction Kinetics Measurements

The thermogravimetric analysis (TGA) of the *MSC* torrefaction was done according to the methodology given by Pulka et al. [38]. To obtain an atmosphere without oxygen, $CO_2$ was injected with a flow rate of 0.6 mL·min$^{-1}$. The tested sample of *MSC* was located inside the cuvette and placed in the center of the reactor. The cuvette containing the sample was combined with a precise (0.01 g resolution) electronic balance for the determination of the mass drop during the process. The readings (temperature, and mass of the sample) were recorded by PC. Dried samples with a mass of 2.25 g were treated under temperatures in the range from 200 to 300 °C (with an interval of 20 °C) for up to 60 min. The chosen temperatures were in the range of the torrefaction previously used by Bialowiec et al. [37]. The torrefaction rate constant (*k*) was determined using the first-order equation [42]:

$$M_s = M_s^0 \times exp^{-k \times t} \tag{1}$$

$$ln M_s^0 / M_s = k \times t \tag{2}$$

where: $M_s^0$ = initial *MSC* mass, g, $M_s$ = mass after time *t*, g, *k* = torrefaction rate constant, s$^{-1}$, *t* = time, s.

The Arrhenius equation expresses the relationship between *k* and temperature (*T*):

$$k(T) = Aexp - E_a / RT \tag{3}$$

and it may be given in the logarithmic form:

$$ln k(T) = ln A - E_a / RT \tag{4}$$

where: *A* = frequency factor; $E_a$ = activation energy; J·mol$^{-1}$; *R* = gas constant, 8.314, J·(mol·K)$^{-1}$; *T* = temperature, K.

$E_a$ was determined with the application of *k* and the Arrhenius equation, with the assumption that *ln*(*k*) is a linear function of 1/*T*:

$$y = ax + b \tag{5}$$

where: $y = ln(k)$, $b = ln A$, $a = E_a / R$.

*2.3. MSC Torrefaction and Torrefied Biomass Generation*

*MSC* torrefaction was executed in accordance with Syguła et al. [43] in a muffle furnace, which is capable of maintaining heating rate, the desired temperature for the desired time, and has a current temperature indication (thermocouple) inside the furnace (SNOL 8.1/1100, Utena, Lithuania). $CO_2$ with the flow rate of 10 $dm^3 \cdot h^{-1}$ was used to establish oxygen-free conditions. The process was carried out under temperatures from 200 °C to 300 °C with intervals of 20 °C. For each temperature, torrefaction was carried out with 20, 40, and 60 min retention time. The samples were heated with a heating rate of 50 °C·$min^{-1}$ from 20 °C to the torrefaction setpoint temperature. Ten grams (± 0.5 g) of the dry mass of the sample was used for the tests. The biocoals were removed from the muffle furnace when the interior temperature was lower than 200 °C (temperature was monitored by an internal thermocouple and visualized on the screen of the furnace). The approximate times of cooling from 300 °C, 280 °C, 260 °C, 240 °C, and 220 °C to 200 °C were 38, 33, 29, 23, and 13.5 min, respectively. The approximate cooling time from 300 °C to room temperature (~20 °C) was around 6 h. Analyses of three replicates were carried out.

*Mass yield*, *energy densification ratio*, and *energy yield* were determined based on Equations (6)–(8) [44].

$$Mass\ yield = \frac{Mass\ of\ dry\ biochar}{Mass\ of\ dried\ raw\ MSC} \cdot 100 \tag{6}$$

$$Energy\ densification\ ratio = \frac{HHV\ of\ biochar}{HHV\ of\ raw\ MSC} \tag{7}$$

$$Energy\ yield = mass\ yield \cdot energy\ densification\ ratio \tag{8}$$

where:

*Mass of dry biocoal*—mass of (dry) biocoal after the process of torrefaction, g;
*Mass of dried raw MSC*—dried mass of *MSC* used in the process of torrefaction, g;
100—conversion to percent;
*HHV of biocoal*—higher heating value of biocoal after the process of torrefaction, $J \cdot g^{-1}$;
*HHV of raw MSC*—higher heating value of dried *MSC* (raw material) used for torrefaction, $J \cdot g^{-1}$.

*2.4. Proximate Analysis*

The samples were tested in three replicates for:

- Moisture content, determined in accordance with [45], by means of the lab dryer (WAMED, KBC-65W, Warsaw, Poland) (Raw *MSC* and biocoals).
- Volatile solids, determined in accordance with [46], by means of the SNOL 8.1/1100 lab muffle furnace (Utena, Lithuania) (Raw *MSC*).
- Ash content, determined in accordance with [47], by means of the SNOL 8.1/1100 lab muffle furnace (Utena, Lithuania) (Raw *MSC*).
- High heating value (*HHV*), determined in accordance with [48], by means of the IKA C2000 calorimeter (Raw *MSC* and biocoals).

The obtained data are presented in our previous article [43] and detailed in the Supplementary Material file "*MSC* torrefaction data.xlsx". The data article contains the results of the raw *MSC* and obtained biocoal's fuel properties and TGA analysis.

*2.5. Torrefaction Process Models*

Polynomial functions describing the influence of torrefaction temperature and its duration on mass yield and energy yield of the torrefaction process and the higher heating value of biocoal were built using the raw data [43]. The model parameters were estimated by nonlinear regression analysis.

Regression analysis used a two-degree polynomial with a general form, with intercept ($a_1$) and five regression coefficients ($a_{2–6}$) (Equation (9)).

$$f(T,t) = a_1 + a_2 \cdot T + a_3 \cdot T^2 + a_4 \cdot t + a_5 \cdot t^2 + a_6 \cdot T \cdot t \tag{9}$$

where:

$f$ ($T,t$)—the biocoal property obtained under $T$—temperature, and $t$—residence time conditions,
$a_1$—intercept (-),
$a_{2-6}$—regression coefficient (-),
$T$—temperature, $T$ = 200–300 °C,
$t$—residence time, $t$ = 0–60 min.

The regression analysis was done with the software Statistica 13 (StatSoft, Inc., TIBCO Software Inc. Palo Alto, CA, USA). For the determination of model parameters and the degree of matching to raw data, the determination coefficient ($R^2$) was calculated.

*2.6. Torrefaction Mass and Energy Balance Evaluation*

Based on the results and using the best variant ($T$ and $t$), a simple model [49] was proposed for estimating the possibility of obtaining the heat self-sufficiency of biocoal production. The model also calculates the theoretical mass yield after biocoal recirculation of the torrefied biomass to maintain the torrefaction process itself (biocoal recirculation—Figure 1).
Data for calculations:

- mass of *MSC*, Mg, assumed 1 Mg,
- moisture content of *MSC*, %, assumed 65.32% (Table 2),
- torrefaction parameters of temperature and time assumed 280 °C and 60 min, respectively.

*Main properties of torrefied biomass calculations:*
Mass yield (*MY*) of torrefaction was used for calculation of mass of torrefied biomass after torrefaction:

$$mtb = mr_d \cdot MY \tag{10}$$

where:

*mtb*—mass of torrefied *MSC* after torrefaction process at $T$, and $t$ conditions, kg, *MY*—mass yield of biocoal, %/%, $mr_d$—dry mass of *MSC*, kg.

*HHV* of torrefied *MSC* was based on Figure 2. The chemical energy in torrefied *MSC*:

$$E_{tb} = mtb \cdot HHV \tag{11}$$

where:

$E_{tb}$—chemical energy in torrefied biomass, kJ.

*Heat need for torrefaction process:*
Data for calculations [49]:

- $T_a$—ambient temperature, °C, assumed 15 °C,
- $T_b$—boiling point of water, 100 °C,
- latent heat of the vaporization of water, 2500 kJ·kg$^{-1}$ [50],
- water specific heat, 4.18 kJ·kg$^{-1}$ [51],

- wood specific heat (assumed for these calculations), kJ·kg$^{-1}$, assumed 1.6 kJ·kg$^{-1}$ [51].

  The heat needed to heat water contained in *MSC*:

$$E_w = m_w \cdot Cp_{water} \cdot (T_b - T_a)$$ (12)

 where:

$E_w$—heat needed to heat water contained in *MSC*, kJ
$Cp_{water}$—water specific heat, 4.18 kJ·kg$^{-1}$,
$m_w$—mass of water in *MSC*, Mg.

  The heat needed to water vaporization:

$$E_{ev} = m_w \cdot L_h$$ (13)

where:

$E_{ev}$—heat needed to vaporization of water contained in *MSC*, kJ
$L_h$—latent heat of water vaporization, kJ·kg$^{-1}$.

  The heat needed to heat *MSC* during torrefaction:

$$E_{hw} = mr_d \cdot Cp_{wood} \cdot (T - T_a)$$ (14)

where:

$E_{hw}$—heat needed to heat *MSC* from ambient to torrefaction temperature, kJ
$Cp_{wood}$—specific heat of wood (assumed for these calculations), kJ·kg$^{-1}$.

  Total heat needed to torrefied 1 MG of *MSC*:

$$E = E_w + E_{ev} + E_{hw}$$ (15)

 where:

$E$—heat needed to torrefied *MSC*

*Estimation of the biocoal recirculation rate to obtain heat self-sufficiency of* MSC *torrefaction*
It has been assumed that the boiler and torrefaction unit heat exchange efficiency is 80%. Therefore, the practical heat demand for torrefaction of 1 Mg of *MSC* $E_p$ may be calculated:

$$E_p = \frac{E \cdot 100\%}{80\%}$$ (16)

Therefore, the biocoal recirculation rate should be:

$$\mu_e = \frac{E_p}{E_{tb}}$$ (17)

where:

$\mu e$—the biocoal recirculation rate to obtain the heat self-sufficiency.

## 3. Results

The *k* values significantly ($p < 0.05$) increased with the temperature of the torrefaction process. The lowest *k* value was recorded at 200 °C ($k = 1.7 \times 10^{-5}$ s$^{-1}$) and the highest ($k = 4.6 \times 10^{-5}$ s$^{-1}$) at 300 °C (Table 3). Similar dependence was confirmed by Dhanavath et al. [52]. However, the increase in *k* in relation to torrefaction temperature was not linear. Significant ($p < 0.05$) differences between the *k* values observed for 300 °C and the values obtained in temperatures between 200 and 260 °C; however, no significant increase between 200 °C and 260 °C was observed (Table A1). Torrefaction at 280 °C caused a significant increase of the *k* value in comparison to 200 °C. The results indicate that the temperature must be >280 °C to achieve significant acceleration of the torrefaction process.

**Table 3.** The kinetic parameters of *MSC* torrefaction: torrefaction constant rate (*k*) for given temperatures (with values of determination coefficients (R$^2$) for *k* estimation), Equation (5) parameters, determination coefficient of Equation (5) parameters estimation, and activation energy of *MSC* torrefaction.

| Constant Rate | Temperature, °C | | | | | |
|---|---|---|---|---|---|---|
| | **200** | **220** | **240** | **260** | **280** | **300** |
| $k_1$, s$^{-1}$ | 0.000008 (0.902) | 0.000020 (0.847) | 0.000024 (0.821) | 0.000028 (0.914) | 0.000030 (0.930) | 0.000047 (0.845) |
| $k_2$, s$^{-1}$ | 0.000020 (0.891) | 0.000018 (0.823) | 0.000023 (0.901) | 0.000023 (0.953) | 0.000032 (0.957) | 0.000039 (0.942) |
| $k_3$, s$^{-1}$ | 0.000022 (0.812) | 0.000017 (0.802) | 0.000012 (0.883) | 0.000024 (0.925) | 0.000034 (0.947) | 0.000052 (0.870) |
| Mean ± standard deviation, s$^{-1}$ | 0.000017 ± $7.5 \times 10^{-6}$ | 0.000018 ± $1.5 \times 10^{-6}$ | 0.000020 ± $6.5 \times 10^{-6}$ | 0.000025 ± $2.7 \times 10^{-6}$ | 0.000032 ± $1.7 \times 10^{-6}$ | 0.000046 ± $6.3 \times 10^{-6}$ |
| Equation (5) parameters | | | | | $y = -2678.7 \times -5.5$ | |
| Determination coefficient (R$^2$) of Equation (5) parameters estimation | | | | | 0.893 | |
| Activation energy, J.mol$^{-1}$, | | | | | 22271.4 | |

Similar to [53,54], we estimated the activation energy for the whole torrefaction process. On the base of estimated *k* values, the activation energy of *MSC* torrefaction, reaching about 22.3 kJ·mol$^{-1}$, was determined (Table 3). The obtained $E_a$ value is low in comparison with typical woody biomass, where, for example, for beech and spruce, these values are 150 and 155 kJ·mol$^{-1}$, respectively [55], for willow from 46 to 152 kJ·mol$^{-1}$, [56] pine 131 kJ·mol$^{-1}$, and fir 128 kJ·mol$^{-1}$ [57]. However, these values have been determined by different methods and models. The model used in the present work had a global first-order character, as it was part of the preliminary study. This model allows estimating solid mass yield at a specific temperature and time of the process. Because of the high ash content in *MSC* and the narrow range of used temperatures, we recommend additional experiments on the kinetics of the process, using more complex models based on first-order kinetics with distributed activation energy models (DAEMs), pseudokinetics, or multicomponent first-order kinetic models.

The experiment showed that the *HHV* increased due to torrefaction in all studied variants in comparison to raw *MSC* biomass (13.7 MJ·kg$^{-1}$ d.m.) (Figure 2). The significantly ($p < 0.05$) lowest values were noted for temperature 200 °C (all durations) from 14.1 to 14.4 MJ·kg$^{-1}$ d.m. The best ($p < 0.05$) fuel properties were noted for biocoals obtained from *MSC* under 280 °C 16.9–17.9 MJ·kg$^{-1}$ d.m. In the case of 220 °C, a significant ($p < 0.05$) influence of torrefaction duration on the increase of *HHV* was noted (Table A2). On the other hand, the increase of torrefaction time decreased ($p < 0.05$) the *HHV* under the temperature of 300 °C, which could be related to organic matter volatilization and mineralization. Additionally, the increase of the temperature from 200 to 280 °C, for the same duration, increased ($p < 0.05$) the *HHV* (Figure 2, Table A2). The experiment showed that to achieve the highest *HHV* of the biocoal, *MSC* should be torrefied under the temperature of 280 °C for a duration of 60 min.

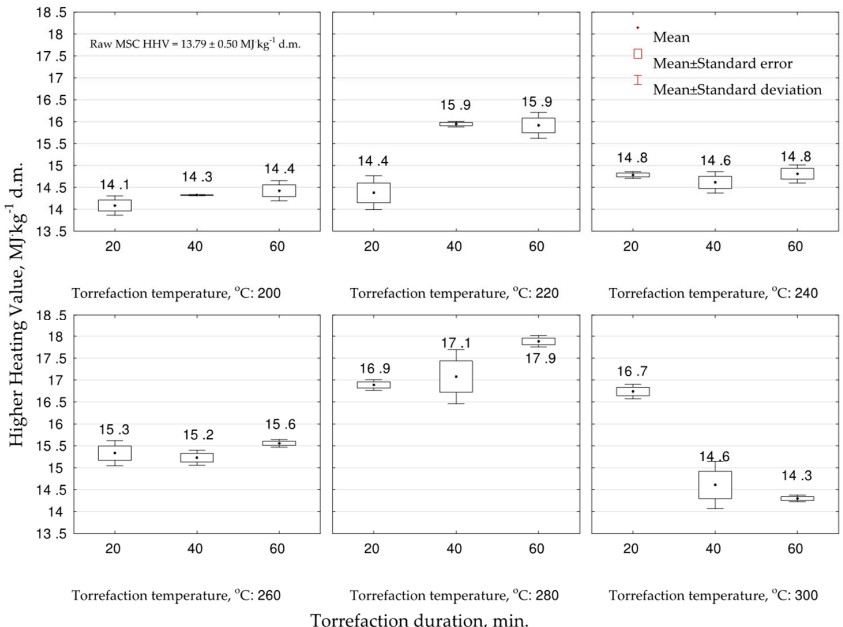

**Figure 2.** The mean higher heating value of raw *MSC* and biocoals produced from *MSC* in relation to torrefaction temperature and torrefaction duration.

The second-degree polynomial model of torrefaction temperature and duration influence on *HHV* was proposed (Figure 3) using raw data presented previously [43]. The statistical evaluation showed that the determination coefficient was relatively low (0.357) (Figure 3) and only two model regression coefficients ($a_3$, and $a_6$) were statistically significant ($p < 0.05$) (Table A3). Such a low fitting degree of the model parameters to existing data could be due to a relatively low number of repetitions (n = 3 for each variant) and/or a low degree of the polynomial. To achieve better results suitable for *MSC* torrefaction optimization, more sophisticated research with a higher number of repetitions (to reduce the degree of the heterogeneity of the results) with the application of better-fitting models is required.

Higher Heating Value, MJ·kg$^{-1}$ d.m. =
$$= -9.1877 + 0.1638*T - 0.0003*T^2 + 0.1148*t + 0.0003*t^2 - 0.0005*T*t$$
$$R^2 = 0.357$$

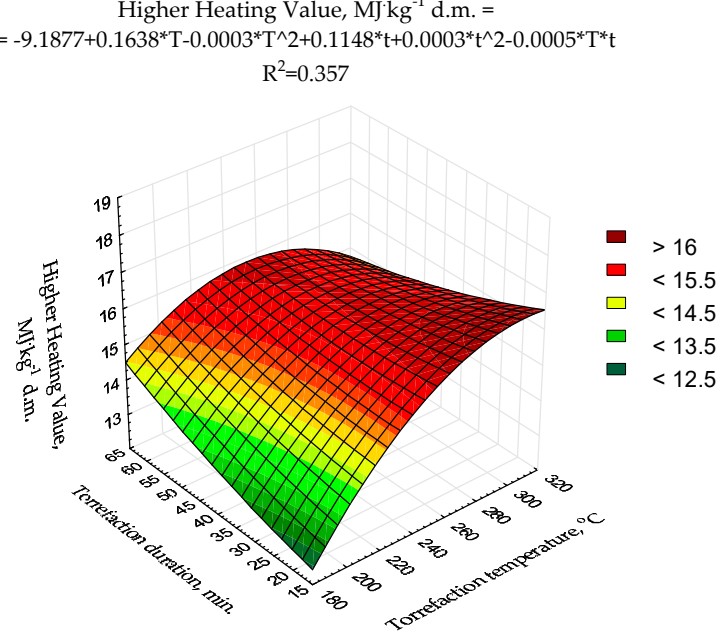

**Figure 3.** The influence of torrefaction temperature and torrefaction time on *HHV* of biocoals produced from *MSC*. The mathematical model (Equation (9)) includes parameters and the determination coefficient of the model's fit to the experimental data ($R^2$).

We determined the mass yields and energy yields to analyze the treatment efficiency. These parameters were calculated according to Equations (6)–(8) on the base of the mean values of *HHV* and masses from the TGA [43] for a given torrefaction temperature and duration. The highest mass yield of biocoal was achieved for temperatures 200 and 220 °C and ranged between 96.9 and 98.5%, respectively (Figure 4). The mass yield under higher temperatures decreased and depended on torrefaction duration. The increase of torrefaction time decreased the mass yield. The lowest achieved value, 87.9%, was in the case of 300 °C, and 60 min duration (Figure 4). For scaling-up the process, high mass yield is important when biomass or waste is converted to biocoal for agricultural purposes. Therefore, for the optimization of that goal, the second-degree polynomial model of torrefaction temperature and duration influence on biocoals' mass yield was proposed (Figure 5). The statistical evaluation showed that the determination coefficient was high (0.953) (Figure 5), but only the model regression coefficients related to torrefaction temperature ($a_2$, $a_3$, and $a_6$) and the intercept ($a_1$) were statistically significant ($p < 0.05$) (Table A4).

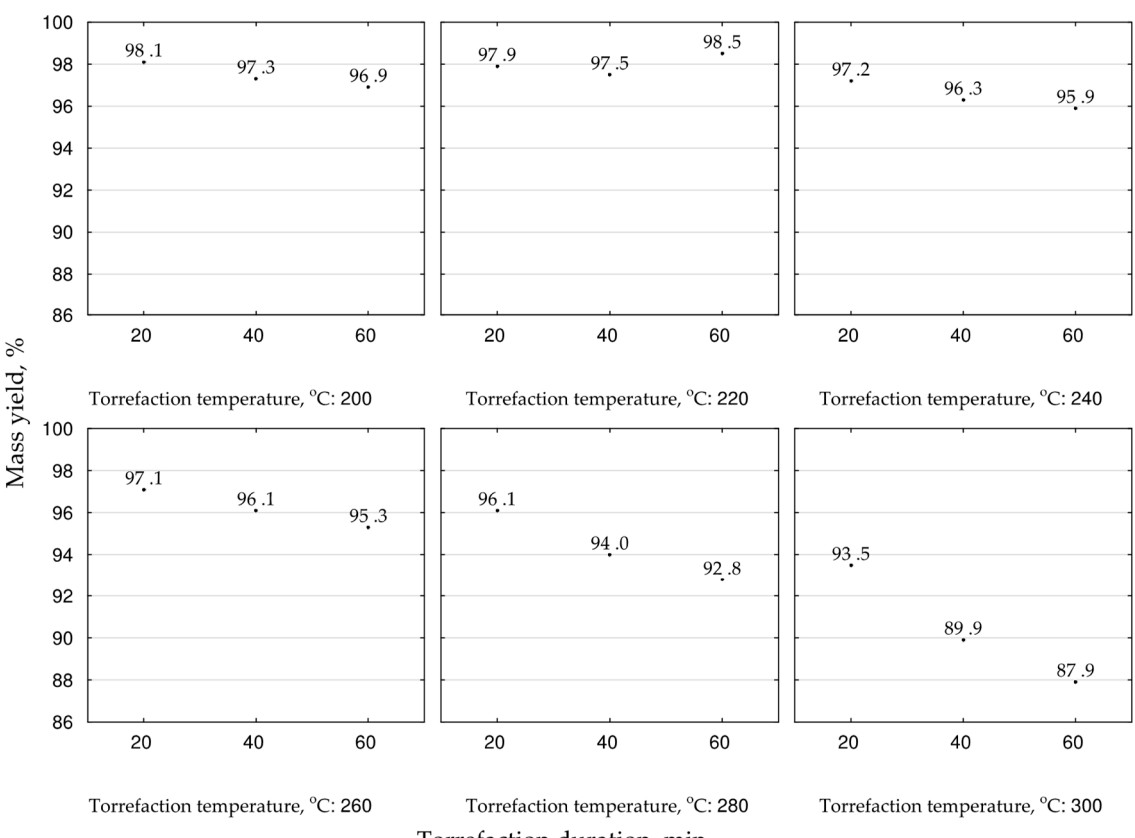

**Figure 4.** The dry matter mass yield values of biocoals produced from *MSC* in relation to torrefaction temperature and torrefaction duration.

The energy yield in biocoal indicates the efficiency of the chemical energy densification due to thermochemical processes concerning the mass loss of the treated *MSC*. The highest energy yield values, exceeding 116.5–120.3%, were noted at 280 °C, and were correlated with process duration. The increase of the temperature to 300 °C decreased the energy yield to 91%. Thus, for fuel production, the temperature of 280 °C and the process duration of 60 min should be applied (Figure 6). Additionally, for optimization of *MSC* conversion to biocoal for energy purposes, the second-degree polynomial model of torrefaction temperature and duration influence on biocoals' energy yield was proposed (Figure 7). The statistical evaluation showed that determination coefficient was not high (0.314) (Figure 7); however, all the regression coefficients were significant ($p < 0.05$) (Table A5). The low degree of model fitting to the obtained data could be related to a similarly low degree of fitting of the model

proposed for the *HHV*. *HHV* values are used for energy yield calculations. Similarly, to achieve better results suitable for *MSC* torrefaction optimization, more complex research with a higher number of repetitions (to reduce the degree of the *HHV* results' heterogeneity) with the application of better-fitting models is required.

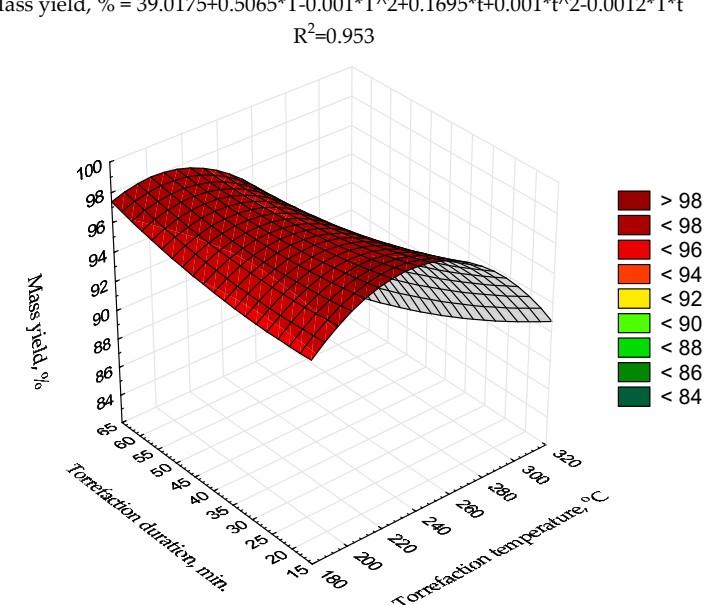

Mass yield, % = 39.0175+0.5065*T-0.001*T^2+0.1695*t+0.001*t^2-0.0012*T*t

$R^2$=0.953

**Figure 5.** The influence of torrefaction temperature and torrefaction time on the dry matter mass yield of biocoals produced from *MSC*. The mathematical model (Equation (9)) includes parameters and the determination coefficient of the model's fit to the experimental data ($R^2$).

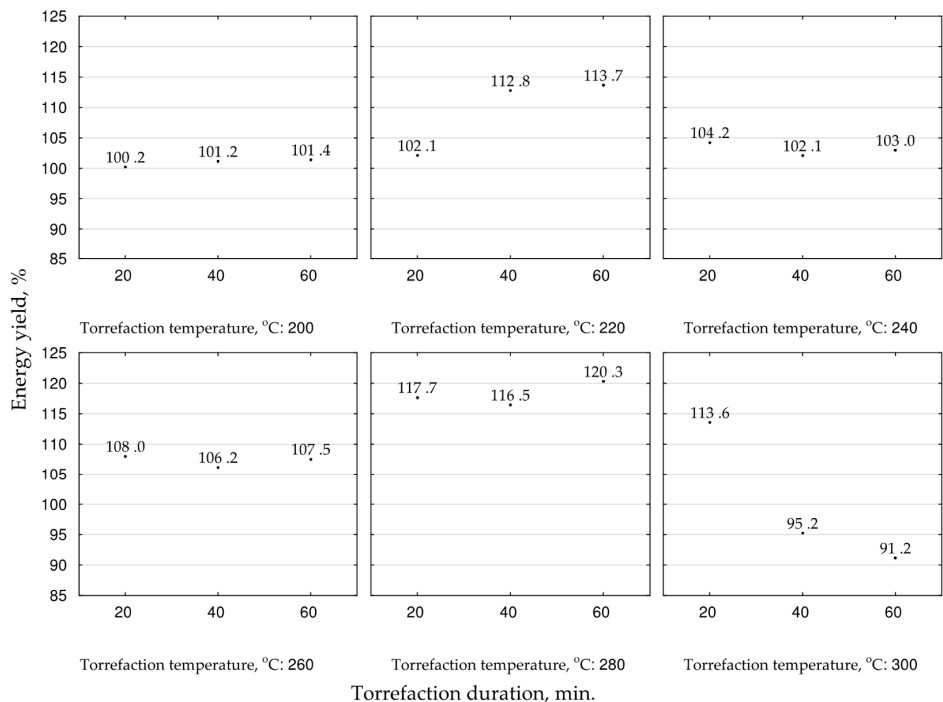

**Figure 6.** The energy yield values of biocoals produced from *MSC* in relation to torrefaction temperature and torrefaction duration.

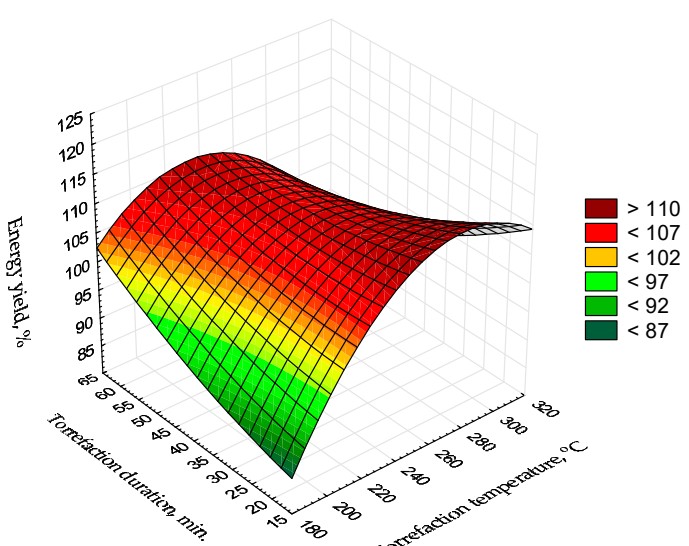

Energy yield, % = -124.8729+1.6815*T-0.0029*T^2+0.9949*t+0.0032*t^2-0.0051*T*t

R$^2$=0.314

**Figure 7.** The influence of torrefaction temperature and torrefaction duration on energy yield in biocoals produced from *MSC*. The mathematical model (Equation (9)) includes parameters and the determination coefficient of the model's fit to the experimental data ($R^2$).

We have shown an alternative utilization of abundant biowaste (*MSC*). *MSC* has potential for improved management that is both sustainable and economical [1–25]. We have proposed treating *MSC* via torrefaction to produce biocoal with improved fuel properties. This research showed that by using torrefaction at 280 °C for 60 min, it is possible to increase the *HHV* of raw *MSC* from ~13.7 MJ·kg$^{-1}$ to ~18 MJ·kg$^{-1}$. This calorific value of obtained biocoal is comparable with nonagglomerating highly volatile coals (*HHV* from 17.4 to 23.9 MJ·kg$^{-1}$) or lignite coal (*HHV* below 17.4 MJ·kg$^{-1}$) [58]. Our research also showed that the highest efficiency (i.e., the energy yield) was achieved at the temperature of 280 °C for a duration of 60 min.

Based on these results and using the best variant (*T* and *t*), a simple model [49] was proposed (after some modifications) for estimating the possibility of achieving heat self-sufficiency of biocoal production. The model also calculates the theoretical mass yield after biocoal recirculation of the torrefied biomass to maintain the torrefaction process itself (biocoal recirculation—Figure 1). The calculations and results were gathered in Table 4.

**Table 4.** Calculation of *MSC* torrefaction mass and energy balance parameters.

| Parameter | Symbol | Calculation | Unit |
|---|---|---|---|
| Mass of torrefied *MSC* after torrefaction process at *T*, and *t* conditions | $m_{tb}$ | 346.8·0.928 = 322.0 | kg |
| Chemical energy in torrefied biomass | $E_{tb}$ | 322.0·17900 = 5763800 | kJ |
| The heat needed to heat water contained in *MSC* | $E_w$ | 653, 2·4.18·(100 − 15) = 232082 | kJ |
| The heat needed to vaporization of water contained in *MSC* | $E_{ev}$ | 653.2·2500 = 1633000 | kJ |
| The heat needed to heat *MSC* from ambient to torrefaction temperature | $E_{hw}$ | 346.8·1.6·(280 − 15) = 147043 | kJ |
| Total heat needed to torrefied 1 Mg of *MSC* | $E$ | 232082 + 1633000 + 147043 = 2012125 | kJ |
| The practical heat demand for torrefaction of 1 Mg of *MSC* | $E_p$ | $\frac{2012125 \cdot 100\%}{80\%} = 2515156$ | kJ |
| The biocoal recirculation rate | $\mu_e$ | $\frac{2515156}{5763800} = 0.436$ | - |

The estimated value of torrefied *MSC* recirculation rate was obtained for 1 Mg of wet *MSC* (65.32% moisture content). For *MSC* torrefaction (280 °C, 60 min), the mass yield of *MSC* biocoal was 92.8%, and *HHV* of torrefied *MSC* was 17.9 MJ·kg$^{-1}$. For these conditions, the estimated biocoal recirculation rate for obtaining the *MSC* torrefaction heat self-sufficiency was 43.6%. Therefore the 56.4% of produced biocoal (181.6 kg produced form 1 Mg of wet *MSC*) may be used for heating the mushroom farm (Figure 1) or be sold to external users.

The presented simplified calculation of the biocoal (considered as a fuel) recirculation rate is the first step for the evaluation of the feasibility of utilization of torrefaction technology for *MSC* treatment. The model was based on simple assumptions. Therefore, more advanced and precise models with a comprehensive analysis of mass and energy balances are warranted as a separate work.

## 4. Conclusions

The presented research revealed that it is possible to produce biocoal from *MSC*. The obtained biocoal has *HHV* similar to a good quality lignite coal. Reaction kinetics analyses of *MSC* torrefaction showed that the intensive organic matter decomposition started above 280 °C. Considering the application of produced biocoal from *MSC* for agriculture, the highest mass yield was obtained under 220 °C. If the *MSC* biocoal production is dedicated to solid fuel production, the torrefaction temperature of 280 °C and process duration of 60 min should be applied to maximize the energy yield. For *MSC* torrefaction (280 °C, 60 min), the mass yield of *MSC* biocoal was 92.8%, and *HHV* of torrefied *MSC* was 17.9 MJ·kg$^{-1}$. For these conditions, the estimated biocoal recirculation rate to obtain *MSC* torrefaction self-sufficiency was 43.6%. The net mass obtained after torrefaction and biocoal recirculation was 0.182 Mg d.m. Second-degree polynomial models for optimization of the torrefaction process were proposed. However, more complex research is required for model calibration and up-scaling. The initial mass and energy balances evaluation showed that *MSC* torrefaction might be self-sufficient for heat, and therefore feasible. This research provides a basis for alternative use of an abundant biowaste and can help chart improved, sustainable mushroom production.

**Supplementary Materials:** The following are available online at http://www.mdpi.com/1996-1073/12/16/3060/s1. Syguła, E.; A. Koziel, J.; Białowiec, A. Waste to Carbon: Preliminary Research on Mushroom Spent Compost Torrefaction. Preprints 2019, 2019060189 (doi: 10.20944/preprints201906.0189.v1). File "*MSC* torrefaction data.xlsx".

**Author Contributions:** Conceptualization, A.B.; methodology, A.B., E.S.; software, E.S.; validation, E.S., A.B., and J.A.K.; formal analysis, E.S.; investigation, E.S.; resources, E.S., A.B., and J.A.K.; data curation, E.S., A.B.; writing—original draft preparation, E.S., A.B.; writing—review and editing, E.S., A.B. and J.A.K.; supervision, A.B and J.A.K.; project administration, A.B.; funding acquisition, A.B., and J.A.K.

**Funding:** The authors would like to thank the Fulbright Foundation for funding the project titled "Research on pollutants emission from Carbonized Refuse-Derived Fuel into the environment," completed at the Iowa State University. In addition, this paper preparation was partially supported by the Iowa Agriculture and Home Economics Experiment Station, Ames, Iowa. Project no. IOW05556 (Future Challenges in Animal Production Systems: Seeking Solutions through Focused Facilitation) sponsored by Hatch Act and State of Iowa funds.

**Conflicts of Interest:** The authors declare no conflict of interest. The funders had no role in the design of the study; in the collection, analyses, or interpretation of data; in the writing of the manuscript; or in the decision to publish the results.

## Appendix A

**Table A1.** Calculated values of probability 'p' of ANOVA analysis with Tukey's Test RSD of variable: torrefaction constant rate (*k*). Highlighted (in bold) differences are significant at $p < 0.05000$.

| Torrefaction Temperature, °C | Calculated Values of Probability '*p*' | | | | |
|---|---|---|---|---|---|
| | **200** | **220** | **240** | **260** | **280** |
| 200 | | | | | |
| 220 | 0.998400 | | | | |
| 240 | 0.976232 | 0.999461 | | | |
| 260 | 0.401048 | 0.618619 | 0.793686 | | |
| 280 | 0.030215 | 0.058894 | 0.099319 | 0.572968 | |
| 300 | 0.000291 | 0.000407 | 0.000562 | 0.003267 | 0.051587 |

**Table A2.** Calculated values of probability '*p*' of ANOVA analysis with Tukey's Test RSD of variable: Higher Heating Value of biocoal. Highlighted differences are significant at $p < 0.05000$.

| Torrefaction Temperature, °C | Torrefaction Duration, min | Calculated Values of Probability '*p*' | | | | | | | | | | | | | | | | |
|---|---|---|---|---|---|---|---|---|---|---|---|---|---|---|---|---|---|---|
| | | 200 20 | 200 40 | 200 60 | 220 20 | 220 40 | 220 60 | 240 20 | 240 40 | 240 60 | 260 20 | 260 40 | 260 60 | 280 20 | 280 40 | 280 60 | 300 20 | 300 40 |
| 200 | 20 | | | | | | | | | | | | | | | | | |
| 200 | 40 | 0.9996 | | | | | | | | | | | | | | | | |
| 200 | 60 | 0.9806 | 1.0000 | | | | | | | | | | | | | | | |
| 220 | 20 | 0.9960 | 1.0000 | 1.0000 | | | | | | | | | | | | | | |
| 220 | 40 | **0.0002** | **0.0002** | **0.0002** | **0.0002** | | | | | | | | | | | | | |
| 220 | 60 | **0.0002** | **0.0002** | **0.0002** | **0.0002** | 1.0000 | | | | | | | | | | | | |
| 240 | 20 | 0.1815 | 0.8098 | 0.9680 | 0.9109 | **0.0009** | **0.0012** | | | | | | | | | | | |
| 240 | 40 | 0.6147 | 0.9965 | 1.0000 | 0.9996 | **0.0002** | **0.0003** | 1.0000 | | | | | | | | | | |
| 240 | 60 | 0.1425 | 0.7419 | 0.9426 | 0.8636 | **0.0012** | **0.0017** | 1.0000 | 1.0000 | | | | | | | | | |
| 260 | 20 | **0.0004** | **0.0057** | **0.0191** | **0.0105** | 0.3675 | 0.4470 | 0.5434 | 0.1461 | 0.6228 | | | | | | | | |
| 260 | 40 | **0.0011** | **0.0202** | 0.0621 | **0.0361** | 0.1546 | 0.2009 | 0.8356 | 0.3518 | 0.8887 | 1.0000 | | | | | | | |
| 260 | 60 | **0.0002** | **0.0004** | **0.0012** | **0.0007** | 0.9386 | 0.9675 | 0.0865 | **0.0124** | 0.1127 | 0.9998 | 0.9862 | | | | | | |
| 280 | 20 | **0.0002** | **0.0002** | **0.0002** | **0.0002** | **0.0125** | **0.0088** | **0.0002** | **0.0002** | **0.0002** | **0.0002** | **0.0002** | **0.0002** | | | | | |
| 280 | 40 | **0.0002** | **0.0002** | **0.0002** | **0.0002** | **0.0012** | **0.0009** | **0.0002** | **0.0002** | **0.0002** | **0.0002** | **0.0002** | **0.0002** | 1.0000 | | | | |
| 280 | 60 | **0.0002** | **0.0002** | **0.0002** | **0.0002** | **0.0002** | **0.0002** | **0.0002** | **0.0002** | **0.0002** | **0.0002** | **0.0002** | **0.0002** | **0.0067** | 0.0632 | | | |
| 300 | 20 | **0.0002** | **0.0002** | **0.0002** | **0.0002** | 0.0694 | 0.0509 | **0.0002** | **0.0002** | **0.0002** | **0.0002** | **0.0002** | **0.0007** | 1.0000 | 0.9799 | **0.0011** | | |
| 300 | 40 | 0.6309 | 0.9972 | 1.0000 | 0.9997 | **0.0002** | **0.0003** | 1.0000 | 1.0000 | 1.0000 | 0.1389 | 0.3382 | **0.0116** | **0.0002** | **0.0002** | **0.0002** | **0.0002** | |
| 300 | 60 | 0.9999 | 1.0000 | 1.0000 | 1.0000 | **0.0002** | **0.0002** | 0.7392 | 0.9909 | 0.6640 | **0.0041** | **0.0146** | **0.0003** | **0.0002** | **0.0002** | **0.0002** | **0.0002** | 0.9924 |

**Table A3.** Statistical parameters of the polynomial model of the influence of torrefaction temperature and process time on the higher heating value of biocoal. Regression analysis used a 2-degree polynomial with a general form, with intercept ($a_1$) and 5 regression coefficients ($a_{2-6}$) (Equation (9)). Highlighted values are significant at $p < 0.05000$.

| Regression Coefficients | Value of Regression Coefficient | Standard Error | T Value, df = 156 | Determined *p*-Value | Lower Range of Confidence Interval | Upper Range of Confidence Interval |
|---|---|---|---|---|---|---|
| $a_1$ (intercept) | −9.18772 | 8.404435 | −1.09320 | 0.279763 | −26.0860 | 7.710533 |
| $a_2$ | −0.00025 | 0.000130 | −1.94593 | 0.057531 | −0.0005 | 0.000008 |
| $a_3$ | **0.16382** | **0.065675** | **2.49436** | **0.016114** | **0.0318** | **0.295867** |
| $a_4$ | 0.00031 | 0.000687 | 0.45333 | 0.652351 | −0.0011 | 0.001692 |
| $a_5$ | 0.11483 | 0.080328 | 1.42947 | 0.159347 | −0.0467 | 0.276336 |
| $a_6$ | **−0.00055** | **0.000232** | **−2.35809** | **0.022491** | **−0.0010** | **−0.000081** |

**Table A4.** Statistical parameters of the polynomial model of the influence of torrefaction temperature and process time on the dry mass yield value of biocoal. Regression analysis used a 2-degree polynomial with a general form, with intercept ($a_1$) and 5 regression coefficients ($a_{2-6}$) (Equation (9)). Highlighted values are significant at $p < 0.05000$.

| Regression Coefficients | Value of Regression Coefficient | Standard Error | T Value, df = 12 | Determined *p*-Value | Lower Range of Confidence Interval | Upper Range of Confidence Interval |
|---|---|---|---|---|---|---|
| $a_1$ (intercept) | **39.01744** | **11.29449** | **3.45455** | **0.004765** | **14.40886** | **63.62603** |
| $a_2$ | **0.50652** | **0.08826** | **5.73904** | **0.000093** | **0.31422** | **0.69882** |
| $a_3$ | **−0.00105** | **0.00017** | **−6.00537** | **0.000062** | **−0.00143** | **−0.00067** |
| $a_4$ | 0.16952 | 0.10795 | 1.57039 | 0.142306 | −0.06568 | 0.40473 |
| $a_5$ | 0.00104 | 0.00092 | 1.12846 | 0.281182 | −0.00097 | 0.00305 |
| $a_6$ | **−0.00122** | **0.00031** | **−3.91408** | **0.002057** | **−0.00190** | **−0.00054** |

**Table A5.** Statistical parameters of the polynomial model of the influence of torrefaction temperature and process time on energy yield value of biocoal. Regression analysis used a 2-degree polynomial with a general form, with intercept ($a_1$) and 5 regression coefficients ($a_{2-6}$) (Equation (9)).

| Regression Coefficients | Value of Regression Coefficient | Standard Error | T Value, df = 12 | Determined *p* Value | Lower Range of Confidence Interval | Upper Range of Confidence Interval |
|---|---|---|---|---|---|---|
| $a_1$ (intercept) | −124.873 | 119.6896 | −1.04331 | 0.317370 | −385.654 | 135.9085 |
| $a_2$ | 1.681 | 0.9353 | 1.79781 | 0.097396 | −0.356 | 3.7193 |
| $a_3$ | −0.003 | 0.0018 | −1.55785 | 0.145239 | −0.007 | 0.0011 |
| $a_4$ | 0.995 | 1.1440 | 0.86967 | 0.401540 | −1.498 | 3.4874 |
| $a5$ | 0.003 | 0.0098 | 0.32266 | 0.752508 | −0.018 | 0.0245 |
| $a6$ | −0.005 | 0.0033 | −1.55290 | 0.146412 | −0.012 | 0.0021 |

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
