# Peer review of "Proof-of-Concept of Spent Mushrooms Compost Torrefaction—Studying the Process Kinetics and the Influence of Temperature and Duration on the Calorific Value of the Produced Biocoal"

_energies, doi:10.3390/en12163060_

Round 1

Reviewer 1 Report

This is a very interesting material, but I have a number of comments. I made 14 comments directly on the text of the manuscript (enclosed). In addition, I would like to mention:

1. You cannot claim on "proof of the concept of a new technology" if the tests were performed on samples of 2.25g and 10g! Samples with a larger mass may behave quite differently, e.g. due to heat transport.

2. Please specify the TRL grade.

3. There were 9 repetitions (as shown in the Supplement tables) in a given temperature and time, however, it seems that it was always the same sample size. It is a pity that this parameter has not been changed to determine the possible impact on kinetics and technical parameters.

4. In point 4. Discussion "a simple model" for estimating the value of biochar produced is presented. If the authors wanted such an assessment, then the rules should be described in p.2 Materials and Methods and the results in 3. Results. According to me, this is a pseudo economic analysis! The authors do not know the definition of "Profit"! Who saw it expressed in Mg !?

"Profit, also called net income, is a financial benefit that is realized when the amount of revenue gained from a  business activity exceeds the expenses, costs, and taxes needed to sustain the activity; money that is earned in trade or business after paying the costs of producing and selling goods and services".

5. I suggest omitting these pseudo-economic analyzes.

6. In order to save space, Figures: 2, 4, 6, should be in the form of three-dimensional bar charts.

7. 12 literature references are to the articles in a non-conference language! They are unsuitable for the worldwide readers and should be replaced with English-language articles.

8. It is advisable to refer to international (ISO, DIN, ASTM) versions of standards.

Author Response

We uploaded our responses in the attached file.

Reviewer 2 Report

This manuscript provides technical data and related analysis on the torrefaction of waste mushroom compost material. The results will be useful.

Line 3:  I think nobody will publish “preliminary studies”. I think you have done some evaluation of the data collected so I suggest that please remove the terms “preliminary studies” from the manuscript title and use better terms.

Line 5:  Please note that “biochar” has a different meaning than “torrefed biomass”. So, please avoid using the word “biochar” in the entire manuscript. You are welcome to use either “torrefied biomass” or “biocoal”. This is an important comment!

Line 25:  Avoid redundancy in “duration time”. Please either use “duration” or “time”, not both. Please follow this suggestion throughout the manuscript.

Line 61:  Avoid redundancy in “several various”. Just use one of the words.

Lines 182-186:  How did you measure the temperature of the torrefied biomass while cooling? Please indicate it.

Figure 3:  The labels on the graph axes are not clear.

Figure 4:  Please indicate whether the mass yield is based on the wet matter basis or dry matter of the initial mass.

Figure 5:  The labels on the graph axes are not clear. Please indicate whether the mass yield is based on the wet matter basis or dry matter of the initial mass.

 Line 411:  The word “profit” should be removed from this heading because you are not talking about financial terms here. Also, it is not clear to me how or where you used this equation 19.

Author Response

(The authors gave the same response as above.)

Reviewer 3 Report

MSC in Table 1 and Raw MSC in Table 2 are confusing.  Raw MSC means the material before composting?  Please explain clearly.

In Table 3, it could be better if a horizontal line is drawn under values of temperatures.   It likes under.

     200              220             240 …

--------------------------------------------------------------

0.000008  0.000020      0.000024 ……..

The reviewer is interested in the following questions.  Please answer if the authors can. 

Who buys the biochar at 132.3 euro/Mg (dm)? Ash generation is quite high at around 30 %.  How they treat the ash?  They means who buys the biochar.  The ash is just going to a dump site? 

Author Response

(The authors gave the same response as above.)
